# Association of chronic neutrophil activation with risk of mortality

**Marc S. Penn** [1,2]*, **Calum MacRae**[3], **Rebecca F. Goldfaden**[4], **Rushab R. Choksi**[4],
**Steven Smith**[5], **David Wrenn**[2], **Mouris X. Saghir**[2], **Andrea B. Klemes**[6]

1 Summa Health Heart and Vascular Institute, Summa Health, Akron, Ohio, United States of America,
2 Quest Center of Excellence for Cardiometabolic Testing at Cleveland HeartLab, Cleveland, Ohio, United
States of America, 3 Department of Medicine, One Brave Idea – American Heart Association, Harvard
Medical School, Boston, Massachusetts, United States of America, 4 East Coast Institute of Research,
Jacksonville, Florida, United States of America, 5 Department of Pharmacotherapy & Translational
Research, College of Pharmacy, University of Florida, Gainesville, Florida, United States of America,
6 MDVIP, Boca Raton, Florida, United States of America

* pennms@summahealth.org

## Abstract

### Background

Levels of free myeloperoxidase (MPO), a cardiovascular risk marker, have been reported to
decline with standard care. Whether such declines signify decreased risk of mortality
remains unknown.

### Design

Cox proportional hazard models were generated using data from a retrospective cohort
study of prospectively collected measures.

### Participants

Patients (3,658) who had MPO measurements and LDL-C $\geq$ 90 mg/dL during 2011–2015
were selected based on a stratified random sampling on MPO risk level. Baseline MPO was
either low (<470 pmol/L), moderate (470–539 pmol/L), or high ($\geq$540 pmol/L).

### Main outcomes and measures

First occurrence of MACE (myocardial infarction, stroke, coronary revascularization, or all-
cause death).

### Results

Mean age was 66.5 years, and 64.7% were women. During a mean 6.5-year follow-up,
crude incidence per 1000 patient years was driven by death. The incidence and all-cause
death was highest for patients with high MPO (21.2; 95% CI, 19.0–23.7), then moderate
(14.6; 95% CI, 11.5–18.5) and low (2.3; 95% CI, 1.2–4.6) MPO. After adjusting for age, sex,
and cardiovascular risk factors, risk of cardiovascular death did not differ significantly
between patients with high and low MPO (HR, 1.57; 95% CI, 0.56–4.39), but patients with

University of Health Sciences, LITHUANIA

**Data Availability Statement:** All relevant data are
within the paper and its Supporting information
files.

**Funding:** Dr. Penn is the founder of Cleveland
HeartLab, Inc and a consulting Medical Director for

Quest Diagnostics Center of Excellence for Cardiometabolic Testing at Cleveland HeartLab. Dr. Saghir is an employee of Quest Diagnostics, including salary and equity holdings. Dr. David Wrenn is an employee of Quest Diagnostics, including salary and equity holdings. Dr. Klemes is the Chief Medical Officer of MDVIP, Inc. The funders had no role in the no role in study design, data collection and analysis, decision to publish, or preparation of the manuscript. Data collection and analyses were performed by Authors who did not work for either of the funders.

**Competing interests:** Dr. Penn is the founder of Cleveland HeartLab, Inc and a consulting Medical Director for Quest Diagnostics Center of Excellence for Cardiometabolic Testing at Cleveland HeartLab. Dr. Saghir is an employee of Quest Diagnostics, including salary and equity holdings. Dr. David Wrenn is an employee of Quest Diagnostics, including salary and equity holdings. Dr. Klemes is the Chief Medical Officer of MDVIP, Inc. This does not alter our adherence to PLOS ONE policies on sharing data and materials.

high MPO had greater risk of non-cardiovascular (HR, 6.15; 95% CI, 2.27–16.64) and all-cause (HR, 3.83; 95% CI, 1.88–7.78) death. During follow-up, a 100 pmol/L decrease in MPO correlated with a 5% reduction in mortality (HR, 0.95; 95% CI, 0.93–0.97) over 5 years.

## Conclusions

Free circulating MPO is a strong marker of risk of mortality. Monitoring changes in MPO levels over time may provide insight into changes in physiology that mark a patient for increased risk of mortality.

## Introduction

Significant progress has been made in the prevention and treatment of coronary artery disease (CAD), owing in large part to statin therapies used to decrease low-density lipoprotein-cholesterol (LDL-C) levels. Clinical studies found decreases in clinical events and mortality in both primary [1] and secondary [2] prevention. Unfortunately, the growing obesity and diabetes epidemics are reversing the decades-long trends of falling morbidity and mortality from CAD achieved with LDL-C lowering as well as smoking cessation efforts. Now, more than 25 years since the landmark 4S trial [3], we see a plateau in our ability to further lower LDL-C levels [4], suggesting that other biomarkers of CAD risk and additional interventions are needed.

The risk factors for CAD and myocardial infarction (MI) are varied and include hyperlipidemia [3], hypertension [5], depression [6,7], periodontal disease [8,9], and acute respiratory viral syndromes [10]. Regardless of the dominant risk factors in an individual, many have suggested that the final common pathway leading to spontaneous thrombosis and MI is a direct consequence of vascular inflammation [10,11]. Ridker and colleagues have shown across multiple studies that high-sensitivity C-reactive protein (hs-CRP) identifies patients with cardiovascular risk [12] even when traditional lipid biomarkers are normal. Moreover, treating high-risk patients who have elevated hsCRP with statins or anti-inflammatory therapies results in an increase in survival [1] and a decrease in cardiovascular events [2].

Free myeloperoxidase (MPO) has been shown to be a marker of cardiovascular risk in multiple biobank studies [13–15]; however, the value of measuring MPO as a prospective marker in an ambulatory population and the potential clinical benefits of lowering MPO remain unstudied. We previously demonstrated that, in a real world self-selected, low-risk, ambulatory population, initially elevated MPO levels in patients with diabetes (21.3%), prediabetes (15.2%), or no diabetes (14%) were reduced to normal in the majority of patients (MPO levels remained elevated in only 6.7%, 4.0%, and 4.0%, respectively) over 4 years [16]. These data indicate that that traditional risk factor-reduction measurescan decrease MPO levels over time [16]. The goal of the current study is to, for the first time, determine if these reductions in MPO levels are associated with decreases in major adverse cardiac events (MACE) and all-cause mortality in a real-world setting.

## Methods

This retrospective cohort study examined associations between MPO levels and the incidence of MACE and all-cause mortality, as well relationships between observed changes in MPO and incidence of MACE and all-cause mortality. The study population comprised patients who were seen at MDVIP offices in which the physicians were willing to complete the necessary

chart reviews and complete case report forms. The total eligible sample size of patients who had annual wellness panels at MDVIP including an MPO level was approximately 120,000 individuals. The study was conducted under an Waiver of Informed Consent by IntegReview (Initial Review January 24, 2019). MDVIP patients agree to have their data used for research studies when they sign up to be cared for by MDVIP practices. All data were obtained by chart review and completion of the case report for each patient by the office where they received care. Forms were sent to the contract research organization (CRO) for inclusion in the secure Excel based database. This study followed the Strengthening the Reporting of Observational Studies in Epidemiology (STROBE) reporting guideline.

## Cohort development

The cohort was developed in 2 phases. In the first (vanguard) phase, we performed simple random sampling of eligible individuals to determine the distribution of MPO values and event rates in the population to power the overall analysis [17,18]. In the second phase, we enrolled additional patients through a stratified random sampling approach, with stratification based on MPO category as high ($\geq$540 pmol/L), moderate (470 to 539 pmol/L), or low (<470 pmol/L). These represent the standard cutoffs used with the MPO test and are based on the 75% (470 pmol/L) and 95% (540 pmol/L) cutoffs of MPO levels in an apparently health population. Based on the vanguard phase and the sample size estimations, we sampled from these strata in a 1:1:3 ratio (<470, 470-<540, $\geq$540 pmol/L), respectively, to enrich the overall sample with additional patients who had high baseline MPO. We based the overall sample size calculation on the analysis of the relationship between change in MPO and MACE/mortality risk. Specifically, we estimated that a total sample size of 3700 would have 80% power at a 2-sided $\alpha = 0.05$ to detect a 2.5% lower hazard (ie, HR of 0.975) for each additional 25 pmol/L reduction in MPO, assuming a standard deviation of 8.5 pmol/L for change in MPO, an $R^2 = 0.55$ between change in MPO and baseline MPO value, and an overall event rate of 10.3% for the primary outcome.

For the vanguard phase, patients of either sex were eligible for enrollment if they were $\geq$18 years old at the time of first MPO measurement in the MDVIP program, had their first MPO measurement between 2011 and 2015, and had an LDL-C >90 mg/dL on the same day as the first MPO measurement. There were no additional inclusion criteria. For phase 2, we applied 1 additional inclusion criterion: availability of a second MPO $\geq$1 year after the first MPO measurement. This difference resulted in a small number of additional patients (enrolled in the vanguard phase, but with only 1 MPO measurement) who were included in analyses focused only on baseline MPO as the exposure but excluded from analyses using change in MPO as the exposure.

## Data collection

Utilizing data provided from Cleveland HeartLab, the CRO determined patient eligibility based on the study inclusion and exclusion criteria and mailed a case report form for each eligible patient to the appropriate MDVIP clinic. The MDVIP clinic was responsible for completing the DCF and returning it to the CRO for data abstraction. The patients for whom a completed case report form was received by the CRO and inputted into Excel based database formed the study cohorts.

For each patient, we collected data on demographics (date of birth, sex); laboratory results from 2011 through 2015 for MPO, LDL-C, hemoglobin $A_{1c}$ (HbA$_{1C}$), and ApoB lipoprotein; occurrences and dates of MACE (MI, stroke, coronary revascularization, all-cause death); history of pre-diabetes and/or diabetes and date of diagnosis; and date of death or last follow-up.

Cause of death was also ascertained, where possible, and categorized as cardiovascular (MI, aortic dissection, arrhythmia, sudden death, stroke but not heart failure, which was listed separately)-related or not.

## Outcomes

The primary outcome was first occurrence of MACE, defined as MI, stroke, or all-cause death. Follow-up for all patients began on the date of the first MPO measurement (for analyses related to baseline MPO) or the second MPO measurement (for analyses related to change in MPO). Follow-up continued until occurrence of an event or, absent an event, the date of last documented follow-up, at which time patients were right-censored.

## Statistical analysis

Patient baseline characteristics were summarized using standard statistics, including mean (standard deviation, SD) for normally distributed continuous variables, median (interquartile range, IQR) for non-normally distributed continuous variables, and n (%) for categorical variables. Incidence rates for the outcomes were calculated as number of events per 1000 person-years. Two sets of analyses were performed, assessing relationships between (1) baseline MPO and risk of adverse events; and (2) change in MPO and risk of adverse events. For each, Cox regression models were developed for time-to-event analyses with baseline MPO or change in MPO as the primary independent variables. All models controlled for age, sex, prior cardiovascular events, baseline $HbA_{1c}$, and baseline LDL-C. Goodness of fit was assessed, including violations of the proportional hazards assumption, through visual inspection and review of Schoenfeld residuals. For the primary models, we modeled MPO values as cubic splines with 3 knots, centered at 540 pmoL/L (for modeling of baseline MPO), or 0 pmol/L (for modeling of change in MPO). In the models for change in MPO, we modelled for absolute change in MPO (pmol/L). We also controlled for baseline MPO (in the 'unadjusted' model), as well as for the above covariates in the adjusted model. Patients in the change in MPO analysis were censored if they had an event before the second MPO. For all Cox models, we report hazard ratio (HR) point estimates with 95% confidence intervals; $P$ values < .05 from the log-rank test were considered statistically significant. Analyses were performed with SAS 9.4 (Cary, NC, USA).

## Results

Phase 1 (vanguard phase) of the study comprised chart review of 766 patients. The goal of phase 1 was to determine event rates and undertake a power analysis for the subsequent broader study. Following analysis of phase 1 (event rate for MACE of 11%), we concluded that we needed a total of 3700 patients. Based on results from the phase 1 analysis, we modified the case report form to include cause of death for phase 2. For the final analysis, case report forms were completed for 3657 patients (766 in phase 1 and 2,891 in phase 2), of whom 3634 had ≥2 MPO measurements. The entire cohort had an event rate of 13.2% during a median follow-up from the initial MPO measurement of 6.5 years (5.8 years from second MPO measurement in the change-in-MPO analysis); the total of follow-up was 22,454 patient years for the initial analyses and 19,101 patient years for the follow-up analyses.

Table 1 lists patient baseline characteristics for the ambulatory population included in the study. Only 23 patients did not have ≥2 MPO measurements; therefore, the patient characteristics are nearly identical between the 2 analyzed cohorts. The mean patient age of the baseline cohort was 66.5 years at first MPO measurement, and most (65%) patients were women. Of note, >90%, of the MDVIP population is Caucasian. Most (64.5%) of the cohort was non-diabetic ($HbA_{1c}$<5.7) 17.7% were pre-diabetic ($HbA_{1c}$ 5.7–6.4% or noted in history), and 17.9%

**Table 1. Characteristics of baseline MPO cohort and "change in MPO cohort" (patients with multiple MPO measures).**

| Variable | Baseline MPO Cohort | Change in MPO Cohort |
|---|---|---|
| Total, n | 3657 | 3634 |
| Age, mean (SD), y | 66.5 ± 13.3 | 67.7 ± 13.2 |
| Sex, No. (%) | | |
| Female | 2366 (64.7) | 2350 (64.7) |
| Male | 1289 (35.2) | 1282 (35.3) |
| Missing | 2 (0.1) | 2 (0.1) |
| **Prior medical history** | | |
| Diabetes status, No. (%) | | |
| Non-diabetic | 2358 (64.5) | 2342 (64.4) |
| Pre-diabetic | 646 (17.7) | 642 (17.7) |
| Diabetes (type 1 or 2) | 653 (17.9) | 650 (17.9) |
| Myocardial infarction | 90 (2.5) | 88 (2.4) |
| Stroke/cerebrovascular accident | 100 (2.7) | 98 (2.7) |
| Prior revascularization | 152 (4.2) | 152 (4.2) |
| **Laboratory data** | | |
| LDL-C, mg/dL, mean (SD) | 126 ± 28 | 126 ± 28 |
| HbA$_{1c}$, %, mean (SD) | 6.0 ± 0.9 | 6.0 ± 0.9 |
| Missing, No. (%) | 79 (2.2) | 2 (0.1) |
| ApoB, mean (SD), mg/dL | 98 ± 19 | 98 ± 19 |
| **MPO category, pmol/L** | | |
| Low (<470 pmol/L), n = 689 | N = 689<br>mean ± SD: 279 ± 84<br>median (IQR): 273 (214, 337) | N = 689<br>mean ± SD: 279 ± 84<br>median (IQR): 273 (214, 337) |
| Moderate (470–539 pmol/L), n = 659 | N = 658<br>mean ± SD: 501 ± 20<br>median (IQR): 498 (483, 518) | N = 654<br>mean ± SD: 501 ± 20<br>median (IQR): 498 (483, 518) |
| High (≥540 pmol/L), n = 2310 | N = 2310<br>mean ± SD: 809 ± 630<br>median (IQR): 628 (575, 750) | N = 2291<br>mean ± SD: 809 ± 631<br>median (IQR): 628 (575, 750) |

Abbreviations: ApoB, apoliprotein B; HbA$_{1c}$, glycated hemoglobin; IQR, interquartile range; LDL-C, low-density lipoprotein-cholesterol; MPO, free myeloperoxidase; SD, standard deviation.

were diabetic (HbA$_{1C}$ >6.4% or noted in history). At baseline, the prevalence of prior cardiovascular events was 2.5% for MI, 2.7% for stroke, and 4.2% for coronary revascularization.

The average and median MPO values for the study are shown in Table 1, along with the distribution of baseline MPO values between low, moderate, and high groups. Despite this being a relatively low-risk population, the mean MPO value at baseline in the study was 653 ± 546 pmol/L, with most patients (63%) being categorized as having a high MPO level. The remainder of the patients were equally split between the moderate (18%) and low (19%) MPO categories.

Table 2 shows crude incidence rates by baseline MPO category. The overall rate of MACE per 1000 patient years was 21.5, with most events (79.1%) being death. For MI, all-cause death, and MACE, incidence increased with increasing MPO category (Table 2).

The Kaplan Meier curves in Fig 1 show that the risk of MI (Fig 1A) was not significantly associated with baseline MPO ($P$ = .12; Unadjusted HR: 3.07 (95% CI, 0.95–9.97)), whereas there was a strong correlation between all-cause death and MPO ($P$ < .001, Unadjusted HR:

**Table 2. Crude incidence rates (95% CI) per 1000 patient years for components of MACE and all-cause death for the entire cohort and by level of MPO elevation at first measure.**

| MPO Category | MI | CVA | Revasc | All-Cause Death | MACE |
|---|---|---|---|---|---|
| All Subjects | 2.4 (1.9–3.2) | 4.6 (3.8–5.6) | 4.1 (3.4–5.1) | 17.0 (15.4–18.8) | 21.5 (19.7–23.5) |
| Low (≤470 pmol/L), n = 689 | 0.9 (0.3–2.7) | 4.4 (2.7–7.3) | 4.7 (2.9–7.7) | 2.3 (1.2–4.6) | 6.5 (4.3–9.9) |
| Moderate (470–539 pmol/L), n = 659 | 2.2 (1.2–4.1) | 3.1 (1.8–5.3) | 3.6 (2.2–5.9) | 14.6 (11.5–18.5) | 18.7 (15.1–23.2) |
| High (≥540 pmol/L), n = 2310 | 2.9 (2.1–3.9) | 5.2 (4.1–6.5) | 4.1 (3.2–5.3) | 21.2 (19.0–23.7) | 25.9 (23.4–28.6) |

Abbreviations: CI, confidence interval; CVA, cerebrovascular accident; Revasc, revascularization; MACE (major adverse cardiac event, defined as MI, CVA, revascularization, or all-cause death) MI, myocardial infarction; MPO, myeloperoxidase.

7.98 (95% CI, 3.95–16.11)) (Fig 1B). No such correlations were observed for CVA (*P* = .18) or revascularization (*P* = .55).

Fig 1 also shows the Kaplan Meier curves for cardiovascular death (Fig 1C) and non-cardiovascular death (Fig 1D). For cardiovascular death, the unadjusted HR was 4.04 (95% CI, 1.47–11.12) for high vs low MPO and 2.44 (95% CI, 0.80–7.46) for moderate vs low MPO. For non-cardiovascular death, the unadjusted HR was 12.03 (95% CI, 4.47–32.38) for high vs low MPO and 8.26 (95% CI, 2.98–22.90) for moderate vs low MPO. After adjustment for age, sex, history of MI, history of CVA, history of coronary revascularization, history of diabetes mellitus,

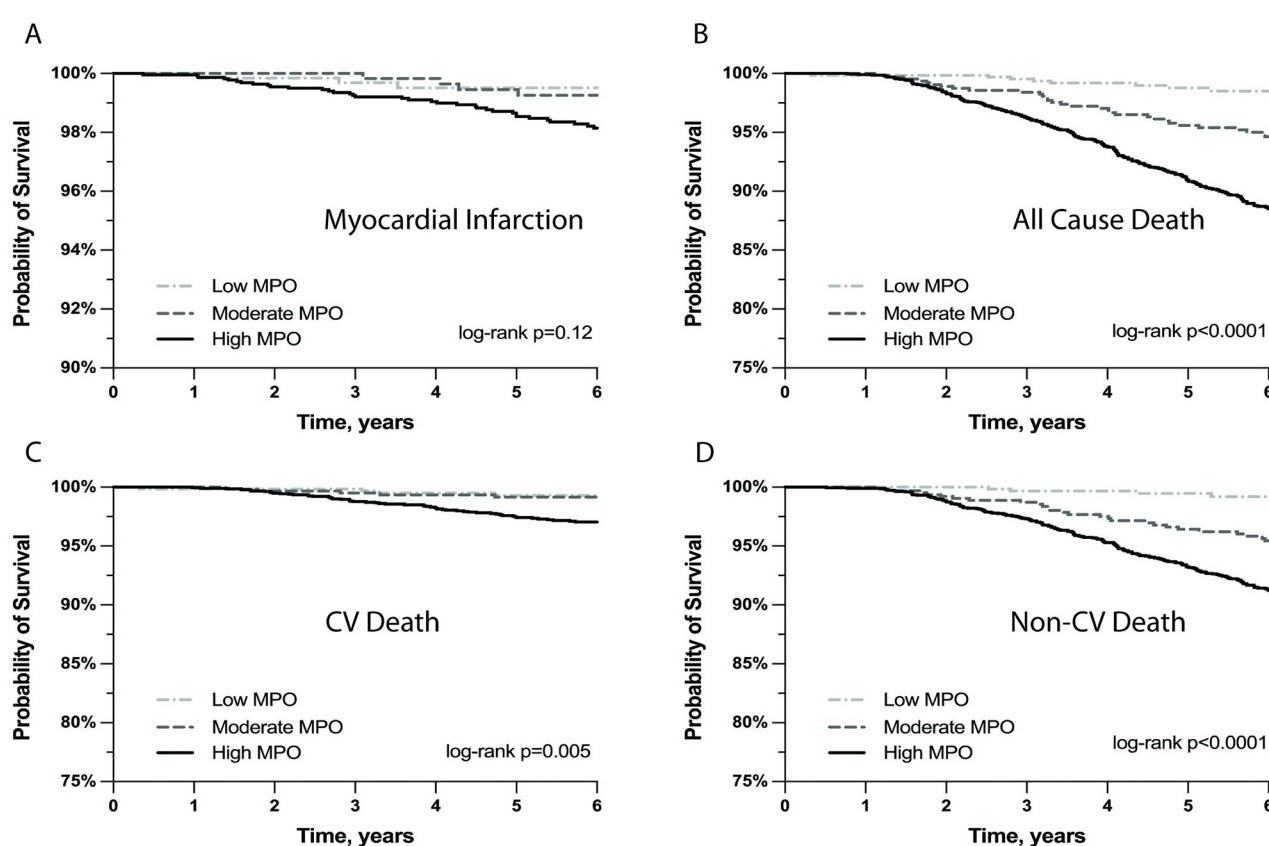

**Fig 1.** Kaplan Meier Curves Depicting Event Rate of (A) Myocardial Infarction, (B) All-Cause Death, (C) Cardiovascular (CV) Death, and (D) Non-CV Death in Patients with an Initial Myeloperoxidase (MPO) Level That Was Low (Light Gray, Dashed Line), Moderately Elevated (Gray, Dashed Line), or Highly Elevated (Black, Solid Line).

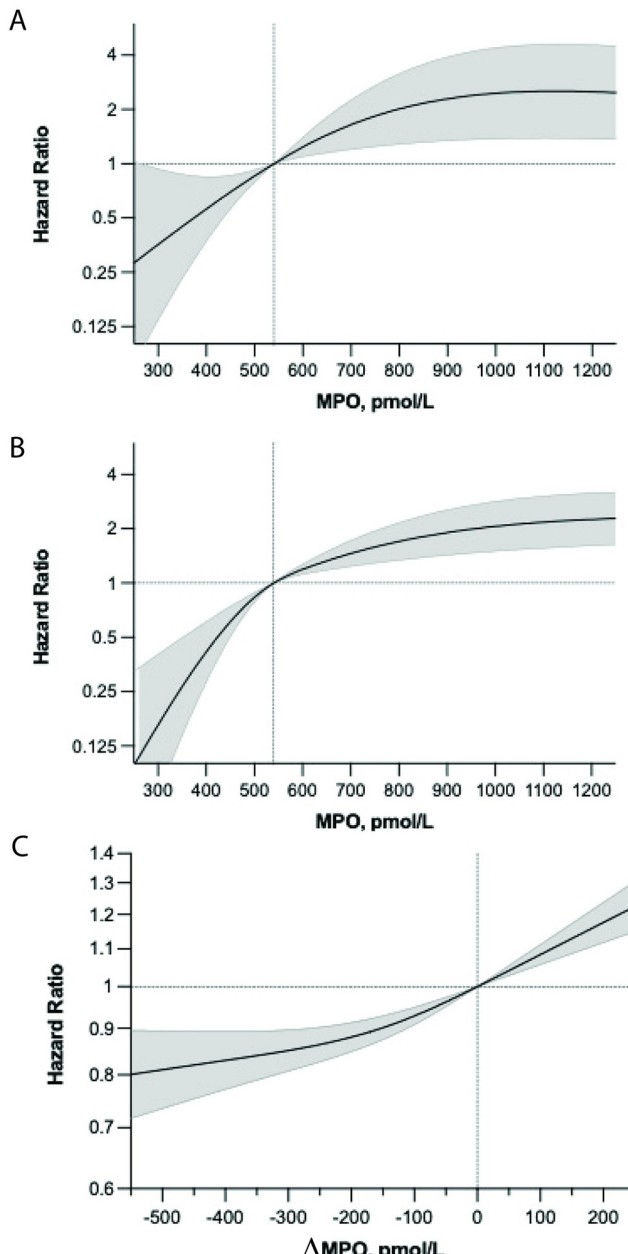

**Fig 2.** Spline Analysis of the Relationship Between Baseline Myeloperoxidase (MPO) Level and the Hazard Ratio of (A) Cardiovascular Death and (B) Non-Cardiovascular Death at 5 Years After Assessment of MPO Level and (C) Change in Hazard Ratio for All Cause Death at 5 Years After Initial Assessment of MPO as a Function of Change in MPO During 5 Years.

baseline LDL-C, and baseline $HbA_{1c}$, the HR for cardiovascular death was 1.57 (95% CI, 0.56, 4.39) for high vs low MPO. For non-cardiovascular death, the adjusted HR was 6.15 (95% CI, 2.27,16.64) for high vs low MPO and 5.04 (95% CI, 1.81,14.02) for moderate vs low MPO.

Fig 2A and 2B demonstrate a continuous, statistically significant relationship between baseline MPO level and risk of cardiovascular and non-cardiovascular death. Fig 2C depicts the relationship between change in MPO after initial MPO and all-cause mortality. These data

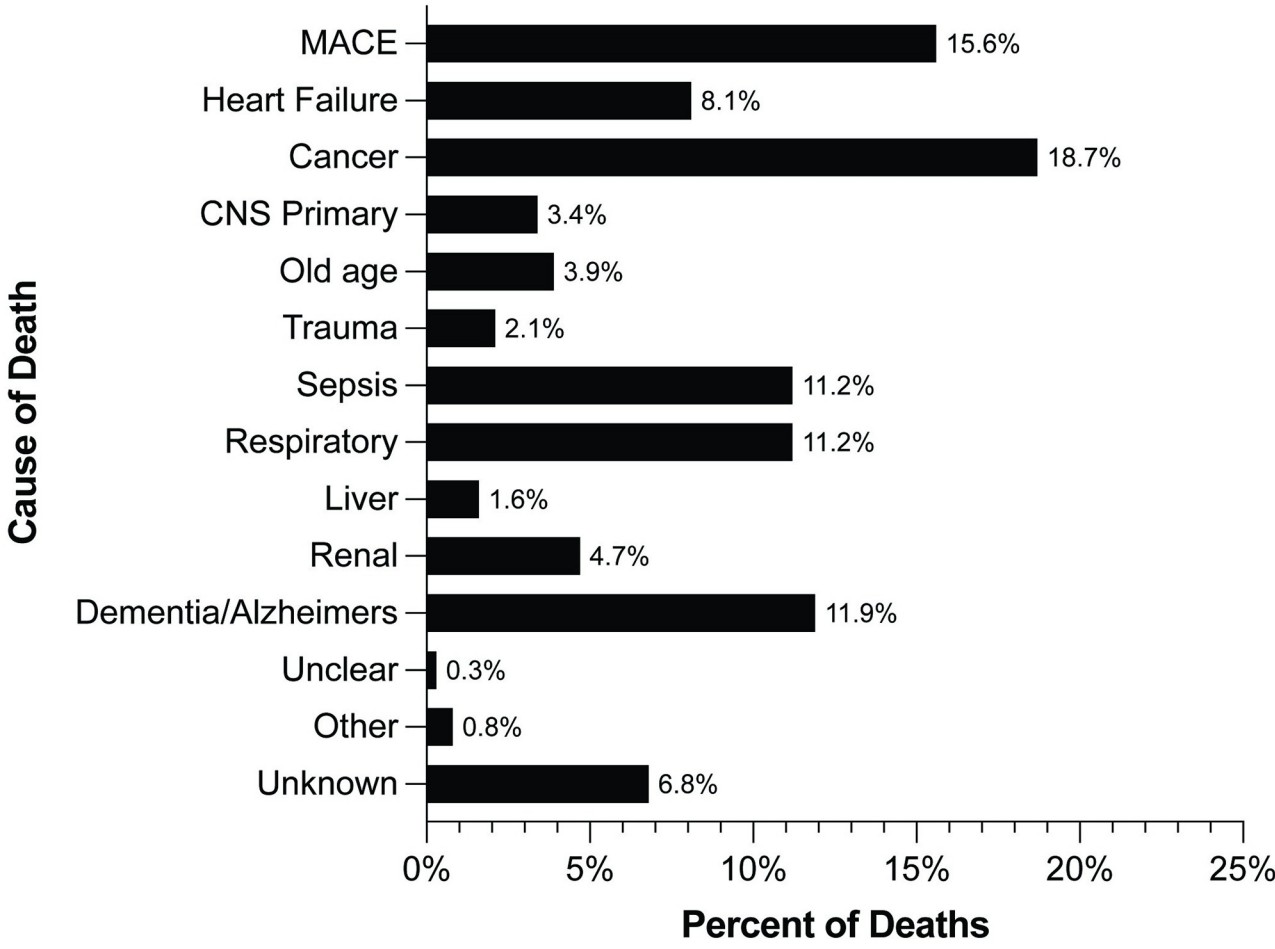

**Fig 3. Causes of death as recorded in medical chart reported as percent of total deaths.**

demonstrate that an initial linear decrease in MPO of up to 100 pmol/L correlates with a significantly lower mortality risk of 5% (HR, 0.95; 95% CI, 0.93–0.97) over 5 years from the initial MPO.

The data in Fig 3 delineate the cause of death based on office chart records. Among a total of 381 deaths a cause of death was able to be determined in all but 25. The combination of heart failure and MI, CVA, revascularization and cardiovascular death accounted for 23.7% of all death. Cancer accounted for 18.7% of the deaths. Other causes of death included liver, renal, respiratory and CNS (Parkinson's and multiple sclerosis specifically) accounted for 20.9% of all deaths. Alzheimer's dementia and the classifier old age (as defined by treating physician) represented an 15.8% of deaths. The largest remaining cause of death accounting for 11.2% of patients was sepsis.

## Discussion

In this retrospective cohort study of prospectively collected measures of MPO in relatively low-risk patients undergoing routine care, free MPO at baseline was a strong marker of subsequent risk of MI, CVA, and death, including cardiovascular, non-cardiovascular, and all-cause death. Moreover, declines in MPO were strongly associated with decreasing risk of adverse

events during follow up. The utility of inflammatory markers in defining cardiovascular is based on the understanding that atherosclerosis has a substantial inflammatory component [11], and the final common pathway for acute arterial events is driven by inflammation. A myriad of risk factors are thought to trigger or exacerbate this inflammation, from depression [6,7] to respiratory viral syndromes [10] and hypercholesterolemia [3].

Numerous inflammatory biomarkers have been studied, including measures of oxidation [19], inflammatory cytokines [20], and proteins including hsCRP [12] and Lp-PLA$_2$ [21,22]. Importantly, there are few data that support these markers as direct signals in the pathophysiology of atherosclerosis [23], but their elevation is associated with poor vascular health. Interestingly, evidence is now emerging that patients with elevated inflammatory markers also have increased risk of cancer [24,25]. The specific relevance of MPO as an inflammatory marker for risk of cancer may be greater than other measures of inflammation given emerging data on the importance of MPO in neutrophil extracellular traps and their role in tumor growth [26,27]. hsCRP was not implemented in MDVIP annual wellness exam due to the groups experience with the non-specificity of the marker. Consistent with that experience, the magnitude of the adjusted hazard ratios of MPO for mortality in this study are greater than literature reports of the adjusted magnitude of the hazard ratio for mortality of 1.3 to 2.2 for hsCRP [28–30]. In this study, we assessed whether reductions in MPO correlated with reductions in MACE and all-cause mortality in a real-world patient population, in which free circulating MPO was prospectively measured. Free MPO is correlates with vulnerable plaque in patients with chest pain [13,15] and has been shown to predict MACE in retrospective analyses of biobank samples in which MPO levels were not a treatment target [14,31]. We previously showed that physician knowledge of MPO levels was associated with a decrease in the prevalence of elevated MPO in subsequent years [16]. The current findings further demonstrate that the decrease in elevated MPO correlated with a decrease in adverse events.

MPO was measured prospectively; therefore, we were able to correlate decreasing MPO with outcomes. Our analysis shows that a decrease in MPO by 100 pmol/L was associated with 5% lower risk of all-cause mortality over 5 years. This finding indicates that reversal of chronic neutrophil activation is associated with improved outcomes. Our data also demonstrate that patients without elevated MPO have lower overall mortality.

Our findings demonstrate that, beyond its potential utility for identifying patients at risk for MI, chronic neutrophil activation as measured by free MPO, identifies patients at risk for mortality in the ensuing 5 years. The degree of risk of death is correlated with the degree of MPO elevation. Over half of the deaths were attributed to either cardiovascular causes or cancer, with the remainder of deaths being associated with conditions of chronic inflammation such as Parkinson's Disease, dementia, non-alcoholic steatohepatitis, or chronic renal failure.

## Limitations

An important limitation to our study is that we were unable to obtain specific drug therapy information on each patient in our cohort. We do know that, in the MDVIP population, the rate of statin use is 60% among patients without CAD or diabetes, compared to ~90% among those with CAD or diabetes. Future studies will need to focus on how elevated MPO levels were addressed.

## Conclusions

Despite these limitations, our study demonstrates that evidence of chronic neutrophil activation as measured by free MPO identifies patients at risk of death. Our data further demonstrate that there is a "dose" effect, in that the risk of adverse events is correlated with the level of

MPO, even in the normal range, and that this risk can be decreased. These findings have implications in both cardiovascular and non-cardiovascular care of patients and suggest that patients who develop elevated MPO warrant greater investigation for potential etiologies that could lead to adverse events. For cardiovascular risk, that could include further risk factor modification personalized to a specific patient's risk factors. For cancer risk, patients with elevated MPO should be encouraged to be up to date on age-appropriate cancer screening. Finally, patients with elevated markers of inflammation, such as MPO, may be appropriate for more advanced screening, such as CT-angiography for cardiovascular risk and liquid biopsy for cancer risk. Conversely, by our data, patients with low MPO appear to be at low risk for adverse events and death over the ensuing 5 years. The dichotomy of outcomes between those with and without elevated markers of inflammation suggests that measures of inflammation could serve to identify patients who need more aggressive investigation and care and could be used to close gaps in care.

## Supporting information

**S1 Data.**
(XLSX)

## Acknowledgments

**Additional Contributions:** East Coast Institute of Research collected all data and performed all statistical analyses.

## Author Contributions

**Conceptualization:** Marc S. Penn, David Wrenn, Mouris X. Saghir, Andrea B. Klemes.

**Data curation:** Marc S. Penn, Calum MacRae, Rebecca F. Goldfaden, Rushab R. Choksi, Steven Smith, David Wrenn, Mouris X. Saghir, Andrea B. Klemes.

**Formal analysis:** Marc S. Penn, Calum MacRae, Steven Smith.

**Funding acquisition:** Mouris X. Saghir, Andrea B. Klemes.

**Investigation:** Marc S. Penn, Rebecca F. Goldfaden, Rushab R. Choksi.

**Methodology:** Marc S. Penn, Rebecca F. Goldfaden, Rushab R. Choksi, David Wrenn, Mouris X. Saghir, Andrea B. Klemes.

**Project administration:** Marc S. Penn.

**Writing – original draft:** Marc S. Penn.

**Writing – review & editing:** Marc S. Penn, Calum MacRae, Rebecca F. Goldfaden, Steven Smith, David Wrenn, Mouris X. Saghir, Andrea B. Klemes.

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
