## [Decision Letter · Decision Letter 0]

29 May 2023

PONE-D-22-14255Association of Chronic Neutrophil Activation With Risk of MortalityPLOS ONE

Dear Dr. Marc S. Penn,

Thank you for submitting your manuscript to PLOS ONE. After careful consideration, we feel that it has merit but does not fully meet PLOS ONE’s publication criteria as it currently stands. Therefore, we invite you to submit a revised version of the manuscript that addresses the points raised during the review process.

We look forward to receiving your revised manuscript.

Kind regards,

Ricardas Radisauskas

Academic Editor

PLOS ONE

3. Thank you for stating the following in the Acknowledgments/ Funding section of your manuscript:

“This study was funded by Quest Diagnostics and MDVIP, Inc”

“Dr. Penn is the founder of Cleveland HeartLab, Inc and a consulting Medical Director for Quest Diagnostics Center of Excellence for Cardiometabolic Testing at Cleveland HeartLab.

Dr. Saghir is an employee of Quest Diagnostics, including salary and equity holdings.

Dr. David Wrenn is an employee of Quest Diagnostics, including salary and equity holdings.

Dr. Klemes is the Chief Medical Officer of MDVIP, Inc.

“Dr. Penn is the founder of Cleveland HeartLab, Inc and a consulting Medical Director for Quest Diagnostics Center of Excellence for Cardiometabolic Testing at Cleveland HeartLab.

Dr. Saghir is an employee of Quest Diagnostics, including salary and equity holdings.

Dr. David Wrenn is an employee of Quest Diagnostics, including salary and equity holdings.

Dr. Klemes is the Chief Medical Officer of MDVIP, Inc.”

6. Thank you for stating the following in the Competing Interests section:

“Dr. Penn is the founder of Cleveland HeartLab, Inc and a consulting Medical Director for Quest Diagnostics Center of Excellence for Cardiometabolic Testing at Cleveland HeartLab.

Dr. Saghir is an employee of Quest Diagnostics, including salary and equity holdings.

Dr. David Wrenn is an employee of Quest Diagnostics, including salary and equity holdings.

Dr. Klemes is the Chief Medical Officer of MDVIP, Inc.”

7. In your Data Availability statement, you have not specified where the minimal data set underlying the results described in your manuscript can be found. PLOS defines a study's minimal data set as the underlying data used to reach the conclusions drawn in the manuscript and any additional data required to replicate the reported study findings in their entirety. All PLOS journals require that the minimal data set be made fully available. For more information about our data policy, please see http://journals.plos.org/plosone/s/data-availability.

8. We note that you have indicated that data from this study are available upon request. PLOS only allows data to be available upon request if there are legal or ethical restrictions on sharing data publicly. For more information on unacceptable data access restrictions, please see http://journals.plos.org/plosone/s/data-availability#loc-unacceptable-data-access-restrictions.

9. We note that you have included the phrase “data not shown” in your manuscript. Unfortunately, this does not meet our data sharing requirements. PLOS does not permit references to inaccessible data. We require that authors provide all relevant data within the paper, Supporting Information files, or in an acceptable, public repository. Please add a citation to support this phrase or upload the data that corresponds with these findings to a stable repository (such as Figshare or Dryad) and provide and URLs, DOIs, or accession numbers that may be used to access these data. Or, if the data are not a core part of the research being presented in your study, we ask that you remove the phrase that refers to these data.

10. Please include your full ethics statement in the ‘Methods’ section of your manuscript file. In your statement, please include the full name of the IRB or ethics committee who approved or waived your study, as well as whether or not you obtained informed written or verbal consent. If consent was waived for your study, please include this information in your statement as well.

Additional Editor Comments:

Dear manuscript authors,

We apologize for the very long manuscript review process, as most of the selected and author-suggested reviewers declined to review.

Your submitted manuscript is suitable for publication in the PlosOne journal with minimal revisions.

Reviewers' comments:

Reviewer's Responses to Questions

**Comments to the Author**

1. Is the manuscript technically sound, and do the data support the conclusions?

Reviewer #1: Yes

Reviewer #2: Yes

2. Has the statistical analysis been performed appropriately and rigorously? 

Reviewer #1: Yes

Reviewer #2: Yes

3. Have the authors made all data underlying the findings in their manuscript fully available?

Reviewer #1: Yes

Reviewer #2: Yes

4. Is the manuscript presented in an intelligible fashion and written in standard English?

Reviewer #1: Yes

Reviewer #2: Yes

5. Review Comments to the Author

Reviewer #1: I recommend modify the title of the manuscript to "Association ... with risk of MACE" because the outcomes in your study includes not only the mortality but also the incidence of myocardial infarction and stroke.

Reviewer #2: #General Comment: Generally, I want to appreciate the authors for undertaking study on such issues in the community. The article is organized and written in good manner. However, I have the following concerns

Abstract

Participant- it is better if sampling method and statistical analysis was mentioned

Conclusion: it says, free circulating MPO is a strong marker of elevated risk of… ? In-complete sentence.

Method

Citation needed for sample size calculation

Which software did you used for data entry?

Did you check the model fitness?

Did you used log rank test for comparison, if so should stated

Result

Why don’t you present the age group in the table rather than repeating mean, already mentioned in narration?

Discussion

The discussion should structured as summarization of the result, it is better if arguing with previous studies. It shouldn’t repeat the result.

6. PLOS authors have the option to publish the peer review history of their article (what does this mean?). If published, this will include your full peer review and any attached files.

Reviewer #1: No

Reviewer #2: No

---

## [Author Response · Author response to Decision Letter 0]

6 Jun 2023

We thank the Reviewers for their careful review of our manuscript. We have tried to address each of the concerns raised below.

Abstract

Participant- it is better if sampling method and statistical analysis was mentioned

As suggested, we have added the sampling method and statistical approach to the abstract in the revised submission.

Conclusion: it says, free circulating MPO is a strong marker of elevated risk of… ? In-complete sentence.

We thank the Reviewer for catching this. We have completed the sentence (added death) in the revised submission.

Method 

Citation needed for sample size calculation

We have added references 17 and 18 that support the sample size calculations that were done 

Which software did you used for data entry? 

The CRO used a secure database that was based on Excel. This has been added to the revised submission.

Did you check the model fitness? Did you use log rank test for comparison, if so should stated.

We did and have edited the revised submission on pages 9 and 10 accordingly. 

Result

Why don’t you present the age group in the table rather than repeating mean, already mentioned in narration?

The paragraph in the results the Reviewer is referring to summarizes Table 1. We feel it is important to highlight specific demographics in the results. We do not feel the repeating of means is inappropriate and Table 1 has more information including the difference in means between the groups and their standard deviations. Therefore, we, respectfully, have chosen not to alter Table 1.

Discussion 

The discussion should structured as summarization of the result, it is better if arguing with previous studies. It shouldn’t repeat the result.

We understand and agree with the Reviewer. The discussion has been heavily edited and any restatements of the results of the study have been removed. We feel the discussion is tighter and flows better and thank the Reviewer for the suggestion.

---

## [Editor Report · Decision Letter 1]

4 Jul 2023

Association of Chronic Neutrophil Activation With Risk of Mortality

PONE-D-22-14255R1

Dear Dr. Marc S. Penn,

We’re pleased to inform you that your manuscript has been judged scientifically suitable for publication and will be formally accepted for publication once it meets all outstanding technical requirements.

Kind regards,

Ricardas Radisauskas

Academic Editor

PLOS ONE
---

## [Editor Report · Acceptance letter]

11 Jul 2023

PONE-D-22-14255R1 

Association of Chronic Neutrophil Activation With Risk of Mortality 

Dear Dr. Penn:

I'm pleased to inform you that your manuscript has been deemed suitable for publication in PLOS ONE. Congratulations! Your manuscript is now with our production department. 

Kind regards, 

on behalf of

Professor Ricardas Radisauskas 

Academic Editor

PLOS ONE